# The Transcription Factor SpoVG Is of Major Importance for Biofilm Formation of *Staphylococcus epidermidis* under In Vitro Conditions, but Dispensable for In Vivo Biofilm Formation

**DOI:** 10.3390/ijms23063255

**Published:** 2022-03-17

**Authors:** Hannah Benthien, Beate Fresenborg, Linda Pätzold, Mohamed Ibrahem Elhawy, Sylvaine Huc-Brandt, Christoph Beisswenger, Gabriela Krasteva-Christ, Sören L. Becker, Virginie Molle, Johannes K. Knobloch, Markus Bischoff

**Affiliations:** 1Institute of Medical Microbiology and Hygiene, Saarland University, 66421 Homburg, Germany; s9habent@stud.uni-saarland.de (H.B.); linda.paetzold@uks.eu (L.P.); mohamed.elhawy@uks.eu (M.I.E.); soeren.becker@uks.eu (S.L.B.); 2Institute for Medical Microbiology, Virology and Hygiene, University Medical Center Hamburg-Eppendorf, 20246 Hamburg, Germany; beate.fresenborg@gmail.com (B.F.); j.knobloch@uke.de (J.K.K.); 3Department of Pathology, Faculty of Veterinary Medicine, Zagazig University, Zagazig 44511, Egypt; 4Institute of Anatomy and Cell Biology, Saarland University, 66424 Homburg, Germany; gabriela.krasteva-christ@uks.eu; 5Laboratory of Pathogen Host Interactions, Université de Montpellier, Centre National de la Recherche Scientifique, UMR 5235, 34095 Montpellier, France; sylvaine.huc-brandt@umontpellier.fr (S.H.-B.); virginie.molle@umontpellier.fr (V.M.); 6Department of Internal Medicine V-Pulmonology, Allergology and Critical Care Medicine, Saarland University, 66421 Homburg, Germany; christoph.beisswenger@uks.eu

**Keywords:** *Staphylococcus epidermidis*, SpoVG, biofilm formation, polysaccharide intercellular adhesin, PIA, *ica*, murine foreign body infection model

## Abstract

*Staphylococcus epidermidis* is a common cause of device related infections on which pathogens form biofilms (i.e., multilayered cell populations embedded in an extracellular matrix). Here, we report that the transcription factor SpoVG is essential for the capacity of *S. epidermidis* to form such biofilms on artificial surfaces under in vitro conditions. Inactivation of *spoVG* in the polysaccharide intercellular adhesin (PIA) producing *S. epidermidis* strain 1457 yielded a mutant that, unlike its parental strain, failed to produce a clear biofilm in a microtiter plate-based static biofilm assay. A decreased biofilm formation capacity was also observed when 1457 Δ*spoVG* cells were co-cultured with polyurethane-based peripheral venous catheter fragments under dynamic conditions, while the *cis*-complemented 1457 Δ*spoVG::spoVG* derivative formed biofilms comparable to the levels seen with the wild-type. Transcriptional studies demonstrated that the deletion of *spoVG* significantly altered the expression of the intercellular adhesion (*ica*) locus by upregulating the transcription of the *ica* operon repressor *icaR* and down-regulating the transcription of *icaADBC*. Electrophoretic mobility shift assays (EMSA) revealed an interaction between SpoVG and the *icaA*-*icaR* intergenic region, suggesting SpoVG to promote biofilm formation of *S. epidermidis* by modulating *ica* expression. However, when mice were challenged with the 1457 Δ*spoVG* mutant in a foreign body infection model, only marginal differences in biomasses produced on the infected catheter fragments between the mutant and the parental strain were observed. These findings suggest that SpoVG is critical for the PIA-dependent biofilm formation of *S. epidermis* under in vitro conditions, but is largely dispensable for biofilm formation of this skin commensal under in vivo conditions.

## 1. Introduction

Implanted medical devices are becoming more and more important in modern medicine. However, at the same time, these implants pose a serious risk for the patient to develop an implantable device related infection, as they may serve as seeding ground for microorganisms to form a multicellular biofilm on these artificial surfaces. Of the nearly 2 million healthcare-associated infections reported by the Centers for Disease Control per year since the beginning of the 21st century, more than 50% can be attributed to indwelling medical devices [1]. *Staphylococcus aureus* and *S. epidermidis* are the most frequently reported bacterial species causing implantable device related infections [2]. Identification of the latter bacterial species—a highly abundant commensal of the human skin flora [3]—in positive blood cultures was considered for a long time primarily as contamination, however, in recent years, *S. epidermidis* has been also recognized as a relevant source for implantable device-related bloodstream infections [4] and late-onset neonatal sepsis [5]. In fact, *S. epidermidis* is nowadays identified as the most important cause of nosocomial catheter related blood stream infections (CRBSI) in numerous countries including Argentina [6], Germany [7], Portugal [8], South Africa [9], Switzerland [10], and the United States [11]. 

One of the reasons for the success of *S. epidermidis* as a nosocomial pathogen lies in its ability to attach to various materials including metal and plastic, to survive on these materials for several days, and to form biofilms on these materials if the environment allows for this (e.g., after the insertion of the implant material into the host). After primary attachment to a surface, the opportunistic pathogen starts to multiply and accumulate within an extracellular matrix (ECM), which is followed by the maturation of the biofilm and finally the dispersal of single cells (reviewed in [12,13]). Biofilm-released cells may spread to other organ systems, and are reported to induce a prompt and stronger inflammatory response than planktonic or biofilm-grown cells of this species [14].

The majority of *S. epidermidis* strains isolated from explanted medical devices produce an extracellular matrix that includes the polysaccharide intercellular adhesin (PIA) as main exopolysaccharide component, a partially deacetylated, positively charged poly-β(1-6)-N-acetylglucosamine, whose synthesis is mediated by the *ica* locus (reviewed in [15]). Beside various environmental stimuli such as temperature, iron- and oxygen limitation, and the amount of available nutrient sources [16,17], PIA-dependent biofilm formation of *S. epidermidis* underlies several internal regulation mechanisms (reviewed in [13]). The alternative transcription factor σ^B^ and the staphylococcal accessory regulator SarA positively affect PIA production by enhancing the transcription of the polycistronic transcript *icaADBC* [18,19], while IcaR, a transcriptional regulator which is also part of the *ica* locus, reduces PIA production by repressing *icaADBC* transcription [20]. Expression of the *ica* locus in *S. epidermidis* is further affected by serine/threonine kinases Stk1 [21], the catabolite control protein A [22], SarX [23], TcaR [24,25], VraSR [26], non-coding RNAs [27,28], and by phase variation via the insertion/excision of the insertion sequence *IS256* into different regions of the *ica* locus [29].

The lack of an apparent σ^B^ binding consensus within the promoter region of the *ica* locus suggests that the stimulating effect of this alternative sigma factor on *icaADBC* transcription is probably of indirect nature. One putative mediator of σ^B^ activity could be SpoVG (aka BarB in *S. epidermidis*), a DNA/RNA-binding protein that is highly conserved in eubacteria [30]. SpoVG is a known regulator of virulence in a number of pathogenic firmicutes such as *Bacillus cereus* [31], *Listeria monocytogenes* [32], and *S. aureus* [33,34,35]. The regulatory protein is also reported to affect biofilm formation in *B. cereus* [31]. Given that σ^B^ controls the transcription of SpoVG in *S. aureus* [33] and that activity of SpoVG is enhanced by Stk1-mediated phosphorylation in the latter species [36], we wondered whether and how SpoVG might affect biofilm formation and *ica* transcription in *S. epidermidis*. Here, we report on the deletion of *spoVG* in the PIA-biofilm producing *S. epidermidis* central venous catheter isolate 1457 [37] and its impact on growth, *ica* transcription and biofilm formation under in vitro and in vivo conditions.

## 2. Results

### 2.1. Genetic Organization of the yabJ-spoVG Locus in S. epidermidis 1457

Genome comparison indicates that the genetic organization of the *purR*-*yabJ-spoVG-glmU* region is highly conserved among the staphylococcal species *S. aureus* and *S. epidermidis* (Figure 1a). Given that *yabJ-spoVG* is co-transcribed in *S. aureus* under the control of a σ^B^-dependent promoter [33], and that the nucleotide sequences of the σ^B^ promoter in front of *yabJ* are identical between *S. aureus* Newman and *S. epidermidis* 1457 (Figure 1b), one can assume that *yabJ* and *spoVG* are also co-transcribed in *S. epidermidis*, asking for a careful deletion strategy that avoids polar effects on *yabJ*.

To confirm that *yabJ* was not affected by our *spoVG* deletion in *S. epidermidis* 1457, we first tested the transcription rates of *yabJ* and *spoVG* by quantitative Real-Time Reverse-Transcription PCR (qRT-PCR) in the strain triplet 1457, 1457 Δ*spoVG*, and 1457 Δ*spoVG::spoVG* (Figure 1c). After 3 h of growth, comparable *yabJ* transcript rates were seen in all three derivatives, while *spoVG* transcripts were missing in the 1457 Δ*spoVG* derivative, but detected at a comparable level in the wild-type and the *cis*-complemented strain. These findings confirmed that *spoVG* was successfully deleted in strain 1457 Δ*spoVG* without altering *yabJ* transcription, and that *cis*-complementation of the *spoVG* deletion with a functional *spoVG* allowed for a complete restoration of *spoVG* transcription in 1457 Δ*spoVG::spoVG*.

### 2.2. Impact of the spoVG Deletion on Growth of S. epidermidis 1457

In order to determine the impact of the *spoVG* deletion on in vitro growth, the strain triplet was first cultured under static conditions in 96-well microtiter plates on an oCelloScope system [39] in Luria-Bertani (LB) broth at 37 °C and 5% CO_2_ (Figure 2). Under these microaerophilic growth conditions, all three strains exhibited comparable growth kinetics (Figure 2a), although 1457 Δ*spoVG* cultures tended to produce slightly higher cell densities after 4 h of growth than wild-type and 1457 Δ*spoVG::spoVG* cultures (Figure 2b), however, this effect was not statistically significant (*p* ≥ 0.275, Kruskal Wallis test and Dunn’s multiple comparisons test).

To test whether SpoVG might render the growth behavior of strain 1457 under more aerobic conditions, growth kinetics of the strain triplet were next analyzed in glass flasks in the bacterial cell culture medium tryptic soy broth (TSB) at 37 °C and 225 rpm (culture to flask volume 1:10). During the early growth stages (i.e., *lag* phase and early exponential growth phase), again no clear differences in growth rates were observed between all three strains (Figure 3a and Table 1).

However, at about 4 h of cultivation, first cell aggregates on the glass surface at the air-liquid interface became visible with the wild-type and 1457 Δ*spoVG::spoVG* cultures but not with the 1457 Δ*spoVG* cultures, suggesting that parts of the wild-type and 1457 Δ*spoVG::spoVG* cell populations started to switch from planktonic growth to a biofilm-mode of growth.

In line with this partial change in growth mode, optical densities of the wild-type and 1457 Δ*spoVG::spoVG* cultures at 600 nm (OD_600_) fell significantly back behind the OD_600_ values seen with 1457 Δ*spoVG* cultures (Figure 3b). This growth behavior was maintained by all three strains until the end of the observation period of 12 h, at which strong biofilms on the glass surface at the air-liquid interface were observed for the wild-type and 1457 Δ*spoVG::spoVG* cultures, which were completely missing with the 1457 Δ*spoVG* cultures (Figure 3c). These findings suggest that SpoVG affects the growth behavior and biofilm formation capacity of *S. epidermidis* strain 1457, if cultured in TSB.

### 2.3. Impact of the spoVG Deletion on Biofilm Formation of S. epidermidis 1457

To substantiate our hypothesis that the *spoVG* deletion in *S. epidermidis* 1457 might affect biofilm formation, three different in vitro biofilm assays were carried out. We first studied the biofilm formation capacity of the strain triplet under static conditions in cell culture-treated 96-well flat-bottom microplates as described in [40]. After 18 h of growth in TSB, cultures of the wild-type and the *cis*-complemented strain produced strong biofilms on the bottom of the microtiter plate wells (Figure 4a). Wells inoculated with the Δ*spoVG* mutant, however, produced after 18 h only weak biofilms that were about one *log* lower than the ones observed with strains 1457 and 1457 Δ*spoVG::spoVG* (Figure 4b). Next, we assayed the biofilm formation of the strain triplet under dynamic conditions in a glass tube assay. Here, after cultivation for 18 h at 37 °C and 150 rpm, cultures of the wild-type and the *cis*-complemented strain produced clearly visible biofilms at the air-liquid interface and the bottom of the glass tubes, while the glass tubes of cultures inoculated with the Δ*spoVG* mutant remained virtually free from cell populations sticking to the glass wall (Figure 4c).

Quantification of the biomasses that attached to the glass walls by safranin staining revealed, in line with the macroscopic observations, about 10-times higher absorption rates at 530 nm (*A*_530_) for the biofilms formed by strains 1457 and 1457 Δ*spoVG::spoVG*, if compared to *A*_530_ values obtained with the isogenic *spoVG* deletion mutant in this assay (Figure 4d). To approach more closely the situation encountered by *S. epidermidis* in the host setting, the biofilm formation capacities of the strain triplet were next tested on peripheral venous catheter (PVC) fragments under dynamic conditions [41]. After five days of cultivation under non-limited nutrient conditions, cultures of strains 1457 and 1457 Δ*spoVG::spoVG* produced macroscopically visible biofilms on the catheter surfaces, while no such biofilm formation was observed with the 1457 Δ*spoVG* cultures (Figure 5a). Quantification of the bacterial cell numbers that attached to the catheter surfaces revealed, in line with the macroscopic observations, about 10-times higher colony forming unit (CFU) rates for the wild-type when compared to the 1457 Δ*spoVG* mutant, and this phenotype was again fully complemented by introducing a functional *spoVG* into the *spoVG* deletion mutant (Figure 5b). Notably, albeit of the fact that no clear biofilm was visible on the outer surfaces of the catheter tubing co-cultured with the 1457 Δ*spoVG* mutant, about 5 × 10^6^ CFU/catheter were detached from the catheter surfaces, indicating that the *spoVG* deletion mutant did not fully lose its ability to attach to the PVC surface.

However, we cannot exclude that at least part of the CFUs obtained from the 1457 Δ*spoVG* co-incubated catheter tubing originated from the bacterial cell population, which grew planktonically in the culture medium (Figure 5c), although catheter fragments were carefully washed prior to the detachment of adherent bacterial cells from the catheter surfaces by sonication. It should be also noted here that CFU determinations of detached biofilms might underestimate the real number of viable cells in these biofilms, as sonication of detached biofilms does not necessarily dissolve all bacterial cells from the ECM and might leave some smaller or even larger cell aggregates [42], which would also yield, similar to individual cells, in single colonies (if cultured on agar plates).

### 2.4. Impact of the spoVG Deletion on ica Transcription of S. epidermidis 1457

*S. epidermidis* strain 1457 is known to form a PIA-dependent biofilm under in vitro conditions [37]. Given the clear impact of the *spoVG* deletion on the in vitro biofilm formation capacities of strain 1457, we assumed that SpoVG might influence PIA production via the transcriptional control of the *ica* locus. To test this hypothesis, we quantified the transcription rates of *icaD*, which is part of the *icaADBC* polycistronic transcript encoding the protein machinery required to synthesize PIA, and of *icaR*, encoding a repressor of *icaADBC* [43], in TSB cultured cells by qRT-PCR (Figure 6). In line with our earlier observations [22], we detected the highest *icaD* transcript rates during the early growth phase (i.e., 3 h of growth) in TSB cultured 1457 cells, underlining the growth phase-dependent transcription of the *icaADBC* polycistronic transcript in glucose-containing medium [44]. Deletion of *spoVG* in strain 1457 strongly reduced the *icaD* transcript rates of the mutant at this early growth stage by a factor of ~20-fold, while *cis*-complementation of the 1457 Δ*spoVG* mutant with a functional *spoVG* created a derivative that produced *icaD* transcript levels comparable to the ones seen with the wild-type (Figure 6a). At the later growth stages monitored (i.e., after 6 and 9 h of growth), all three strains produced comparable *icaD* transcript rates, suggesting that SpoVG affects *ica* expression predominantly during the early growth stages. Notably, deletion of *spoVG* in strain 1457 also rendered the transcription of *icaR* (Figure 6b).

While the wild-type and 1457 Δ*spoVG::spoVG* cell populations produced *icaR* transcript rates that were comparable throughout growth, significantly higher *icaR* transcript rates were observed with the 1457 Δ*spoVG* cell populations at all three time points analyzed, if compared to the transcript levels seen with the wild-type cultures. These findings suggest that SpoVG might augment *icaADBC* transcription via the modulation of *icaR* transcription.

### 2.5. Interaction of SpoVG with the icaA-icaR Intergenic Region in S. epidermidis 1457

Animated by our *icaD*/*icaR* transcriptional findings, we conducted a series of electrophoresis mobility shift assays (EMSAs) to test whether SpoVG might directly interact with the *icaA-icaR* intergenic region (Figure 7).

This 165-bp spanning region harbors the transcriptional start sites (TSS) of *icaADBC* and *icaR*, respectively [46,47], and is a known binding site for regulators such as SarA [19], SarX [23], TcaR [24,25,48], and IcaR itself [45]. Two 80-nt spanning double-stranded DNA probes were generated that covered either the *icaA* or *icaR* TSS (termed *icaA* and *icaR* probe, respectively; Figure 7a). Both double-stranded DNA probes shifted with SpoVG in a dose-dependent manner (Figure 7b,c), while the control dsDNA fragment was not shifted in presence of the HAT-HA-tagged DNA/RNA-binding protein. Shifting of the labeled *icaA* and *icaR* probes was furthermore competed out by adding unlabeled *icaA* and *icaR* probes in excess, respectively, indicating that SpoVG can bind directly to the *icaA* and *icaR* promoter regions in *S. epidermidis.*

### 2.6. Impact of the spoVG Deletion on Infectivity of S. epidermidis 1457 in a Murine Foreign Body Infection Model

Earlier work demonstrated that inactivation of the catabolite control protein A encoding gene *ccpA* in *S. epidermidis* 1457 exerts a strong impact on the in vitro biofilm formation capacity of the mutant, but is negligible for the in vivo biofilm formation of *S. epidermidis* in a murine foreign body infection model [22], indicating that in vitro observed phenotypes are not necessarily predictive for the in vivo situation. Driven by the strong impact of the *spoVG* deletion on the biofilm formation capacity of strain 1457 seen here under in vitro conditions (Figure 4 and Figure 5), we wondered now whether and how this deletion might affect the biofilm formation capacity and infectivity of *S. epidermidis* under in vivo conditions. In order to address this question, we utilized a modified version of the murine foreign body infection model described by Rupp and colleagues [49], in which ~1 cm PVC fragments are inserted into the subcutis of both flanks of a mouse and the catheter lumen subsequently infected with a defined bacterial solution. Mice challenged with the Δ*spoVG* mutant displayed a clear reduction in edema sizes around the implanted catheter fragments, if compared to the edema sizes seen with mice infected with the wild-type (Figure 8a). However, only minor differences in bacterial loads on the implanted catheter fragments and the surrounding tissues were detected between wild-type and Δ*spoVG* mutant challenged mice (Figure 8b,c). Mice infected with the Δ*spoVG* mutant produced more heterogeneous CFU numbers on the catheter tubing than mice infected with the wild-type and the *cis*-complemented derivative (Figure 8b), while in infected tissues surrounding the catheter, rather comparable CFU rates were observed for all three mouse groups (Figure 8c). These findings indicate that SpoVG, albeit of being of major importance for PIA production and biofilm formation of *S. epidermidis* 1457 under in vitro conditions, is largely dispensable for biofilm formation on implantable devices under in vivo conditions but does affect the inflammatory response of the host at the infection site.

### 2.7. Impact of the spoVG Deletion on Chemokine Expression in S. epidermidis 1457 Infected Tissue

To test our hypothesis that SpoVG alters the inflammatory response of the host at the infection site, we assayed the contents of the pro-inflammatory cytokines interleukin-6 (IL-6) and tumor necrosis factor α (TNF-α), and of the neutrophil chemoattractant keratinocyte-derived chemokine (KC) in tissue homogenates obtained from excised tissues surrounding the catheter fragments of *S. epidermidis*-infected mice at day 10 post infection. In line with the augmented edema sizes found around the implanted catheter fragments in mice infected with the wild-type (Figure 8a), we observed significantly increased TNF-α contents in the tissue homogenates obtained from wild-type-infected mice.

Notably, mice infected with strain 1457 contained at this late stage of infection still more than twice as much of TNF-α in the infected tissue as mice infected with the isogenic Δ*spoVG* mutant (Figure 9a). Mice infected with the wild-type strain also possessed higher IL-6 and KC level in the infected tissue than mice infected with the Δ*spoVG* mutant (Figure 9b,c), however, these differences were not statistically significant. In combination with the CFU data obtained with this animal model at the infection sites (Figure 8b,c), these cytokine expression data suggest that SpoVG expressing *S. epidermidis* isolates elicit a long-lasting enhanced inflammatory response at the infection site leading to an increased tissue swelling, albeit without markedly altering the bacterial burden at the infection site.

## 3. Discussion

The ability of *S. epidermidis* to adhere to implanted medical devices and to form biofilms on these artificial surfaces is considered as the primary virulence mechanism of this skin commensal [50]. Here, we demonstrate that the DNA/RNA-binding protein SpoVG strongly affects biofilm formation of the PIA-producing *S. epidermidis* strain 1457 under in vitro conditions, presumably by enhancing the transcription of *icaADBC* via down-regulation of IcaR expression. However, we could not identify a major impact of SpoVG on the in vivo biofilm formation capacity of strain 1457 in a murine foreign body infection model, although *S. epidermidis* cells harboring a functional *spoVG* locus elicited a stronger tissue swelling at the infection site in this model, which was accompanied by enhanced amounts of the pro-inflammatory cytokine TNF-α. These findings concur with earlier observations made with a *ccpA* mutant of this *S. epidermidis* strain, in which a strong effect of the *ccpA* mutation on the in vitro biofilm formation capacity was observed which was not accompanied by a clear effect in the in vivo infection model [22]. Given that inactivation of the *ica* locus in strain 1457 is reported to strongly decrease the biofilm formation capacity of the mutant in vitro [46,51,52], and the bacterial load at the infection site in the murine foreign body infection model [49,53,54], one can assume that the regulatory circuits leading to *ica* expression and PIA production between growth in vitro and in vivo are differing. While the σ^B^-dependent transcription factor SpoVG is crucial for PIA production under in vitro conditions, it seems to be dispensable for PIA production of *S. epidermidis* under in vivo conditions, although we cannot exclude that the *spoVG* deletion mutant of strain 1457 might be able to produce a PIA-independent biofilm in this murine infection model, which yields in comparable CFU rates at the infection sites. Support for the latter hypothesis is given by recent findings demonstrating that host factors can compensate for PIA in cell-cell aggregation and biofilm formation of *S. epidermidis* [55].

Notably, strain 1457 was reported to elicit more severe skin and soft tissue inflammatory changes at the catheter insertion sites than the isogenic *ica*-negative isolate M10 [54], a phenotype that was also observed here with the *spoVG* deletion mutant, suggesting that 1457 Δ*spoVG* might produce lower levels of PIA than the wild-type in this infection model. However, the fact that 1457 cells grown in a PIA-dependent biofilm were reported to decrease the level of pro-inflammatory cytokines in a whole blood assay, if compared to M10 cells grown in a biofilm [52], suggests rather that strain 1457 does not produce high levels of PIA in the murine foreign body infection model, given the elevated TNF-α levels seen with the tissues infected with the wild-type strain.

Earlier studies highlighted the impact of the *sigB* operon on PIA-dependent biofilm formation in *S. epidermidis* [18,56,57] and suggested that the alternative σ-factor σ^B^ is likely to promote biofilm formation of strain 1457 by repressing IcaR expression [18]. However, the lack of an apparent σ^B^ consensus sequence in the *ica* locus suggested that σ^B^ is likely to modulate *ica* expression in an indirect manner [58]. Given that the promoter region of the *yabJ-spoVG* locus in *S. epidermidis* strain 1457 harbors a σ^B^ binding motif that is virtually identical to the σ^B^ binding site of the *yabJ-spoVG* locus of *S. aureus* [33], it is tempting to speculate that the impact of σ^B^ on *icaR* transcription in *S. epidermidis* is, at least in part, mediated via the σ^B^-dependent transcriptional regulator SpoVG. Another factor mediating the σ^B^ effect on *ica* expression might be SarA, which positively affects *icaADBC* transcription [19,58], and is in part controlled by σ^B^ [18], although overexpression of the σ^B^-dependent *sarA* P1 transcript in a 1457 *sigB* mutant was found to have, if at all, only little impact on PIA production [58]. However, regulators such as SarA, SarX, and TcaR, which directly control *icaADBC* transcription [19,23,24,48], might explain the discrepancy noticed in our study that deletion of *spoVG* significantly altered the *icaR* transcription at all three growth phases analyzed, while *icaADBC* transcription was altered significantly only in the early growth stage by the mutation. Another factor contributing to this phenotype might be Stk1, a serine/threonine protein kinase reported to enhance the DNA-binding properties of SpoVG towards its target sequences in *S. aureus* [36]. Experiments are ongoing in our laboratory to address this hypothesis.

## 4. Materials and Methods

### 4.1. Bacterial Strains and Plasmids

The bacterial strains and plasmids used in this study are listed in Table 2. All mutant strains generated for this study were confirmed by assessing gene transcription by quantitative real-time reverse-transcription PCR (qRT-PCR).

### 4.2. Bacterial Growth Conditions

*E. coli* strains were grown at 37 °C and 150 rpm in Luria Bertani (LB) medium (BD, Heidelberg, Germany) or on LB-agar plates supplemented with antibiotics when appropriate. Antibiotics were used at the following concentrations: erythromycin, 10 μg/mL; chloramphenicol, 10 μg/mL; ampicillin, 100 μg/mL. *S. epidermidis* strains were routinely grown in tryptic soy broth (TSB; BD) or on TSB plates containing 1.5% agar supplemented with antibiotics when appropriate. For growth kinetics studies, bacterial cells from overnight cultures were diluted into pre-warmed medium to an optical density at 600 nm (OD_600_) of 0.05. Cell suspensions were then either incubated under static conditions in 96-well microtiter plates at 37 °C and 5% CO_2_ in an oCelloScope (BioSense Solutions, Farum, Denmark), or aerobically in flasks at 37 °C and 225 rpm (flask-to-medium ratio 10:1). Growth kinetics of the static cultures were determined using the Background Corrected Absorption (BCA) algorithm of the UniExplorer software (BioSense Solutions, version 9.0). Growth kinetics of aerated cultures were monitored by taking samples at every hour and measuring the OD_600_ values using a Gene Quant 1300 spectrophotometer (Biochrom, Berlin, Germany). Cell suspensions reaching an OD_600_ > 0.8 were diluted 1:10 or 1:20 with phosphate buffered saline (PBS, pH 7.2), and the measured absorbance values were multiplied with the dilution factor. The generation times of *S. epidermidis* strains cultivated in TSB under aerobic conditions were determined as described in [41].

### 4.3. Mutant Construction

Construction of the 1457 *spoVG* deletion mutant was carried out essentially as described in [18]. DNA fragments covering the regions flanking *spoVG* were amplified by PCR using the Expand High Fidelity PCR system (Roche, Mannheim, Germany) with oligonucleotides shown in Table 3 and *S. epidermidis* 1457 DNA (NCBI Reference Sequence: NZ_CP020463.1) as template. The erythromycin resistance cassette *ermB* was cloned from a transposon TN*917* harboring *S. epidermidis* strain using primer pair *ermB* forward/reverse. Amplified fragments were cloned into the DONR vectors of the Multisite Gateway Three-Fragment Vector Construction Kit according to manufactures instructions (Invitrogen, ThermoFisher Scientific, Schwerte, Germany).

Resulting plasmids pENTRY_*spoVG*_up, pENTRY_*spoVG*_do and pENTRY_*ermB* were then combined with the Gateway destination vector pDEST R4-R3 (Invitrogen) in a MultiSite Gateway^®^ LR recombination reaction to create plasmid pDEST_*spoVG*_ko. The *spoVG* up-*ermB*-*spoVG* do DNA region of the latter plasmid was next PCR amplified with primer pair *spoVG*_ko forward/reverse, the amplification product digested with *Pst*I/*Bam*HI, and ligated into the temperature-sensitive *E. coli/Staphylococcus* shuttle-vector pBT2 [63] digested with the same enzymes. The resulting plasmid pBT2_*spoVG*_KO was propagated in *E. coli* TOP10 (Invitrogen) and subsequently transformed into the restriction-deficient staphylococcal genetic background of *S. aureus* RN4220 by electroporation. pBT2_*spoVG*_KO isolated from this staphylococcal host was then transformed by electroporation into *S. epidermis* mutant M15 [56] and subsequently phage-transduced into the recipient strain, *S. epidermidis* 1457 with phage 71. 1457 cells positive for pBT2_*spoVG*_KO were grown at non-permissive temperatures, and clones were screened for double crossover with an erythromycin-resistant but chloramphenicol-sensitive phenotype. The correct chromosomal insertion of the cassette into the respective mutant was demonstrated by PCR with an *ermB*-specific primer paired with a primer flanking the genetically manipulated site and subsequent sequencing of the resulting fragment (data not shown).

For construction of the *spoVG cis*-complemented derivative of the 1457 ∆*spoVG*, the primer pair MBH603/MBH607 was used to amplify a ~3.0 kb fragment including the *yabJ-spoVG* locus from genomic DNA of the strain 1457. The amplification product was digested with *Sac*I/*Bgl*II and ligated into *Sac*I/*Bgl*II-digested and dephosphorylated *E. coli*-*S. epidermidis* shuttle-plasmid pBASE6 [62] to produce plasmid pBASE6_*spoVG*_comp. The temperature-sensitive plasmid was propagated in *E. coli* strain IM08B [61] and plasmids isolated from this *E. coli* strain were directly electroporated into 1457 ∆*spoVG* and cultured at 30 °C on TSA plates supplemented with chloramphenicol (10 µg/mL). Replacement of the *ermB*-tagged *spoVG* deletion by the functional *yabJ-spoVG* locus was carried out as previously described [64].

### 4.4. RNA Isolation and Purification, cDNA Synthesis and qRT-PCR

*S. epidermidis* strains were cultivated in TSB at 37 °C and 225 rpm. Bacterial pellets were collected after 3 h, 6 h, and 9 h of incubation by centrifugation at 5000 rpm at 4 °C for 5 min, and immediately resuspended in 100 µL ice-cold TE-buffer (10 mM Tris-HCl, 1 mM EDTA, pH 8). Bacteria were disrupted, total RNA isolated, transcribed into cDNA, and qRT-PCRs carried out essentially as described previously [65] using the primers listed in Table 3. Transcriptional levels of target genes were normalized against the mRNA concentration of the housekeeping gene *gyrB* according to the 2^−∆CT^ method [66].

### 4.5. Biofilm Assays

The impact of the *spoVG* deletion in strain 1457 on in vitro biofilm formation was assayed in different biofilm models under static and shaking conditions.

Biofilm formation under static conditions was assessed as described in [40]. Briefly, overnight cultures were diluted to an OD_600_ of 0.05 in fresh TSB medium, and 200 µL of the cell suspension was used per well to inoculate sterile Nunc™ MicroWell™ 96-Well, Nunclon Delta-Treated, Flat-Bottom Microplates (ThermoFisher). After incubation for 18 h at 37 °C without shaking, the plate wells were washed thrice with PBS and dried at 65 °C in an inverted position. To apply comparable shear forces to the biofilms, supernatants/wash solutions were removed with the help of a glass capillary attached to a vacuum pump system (KNF, Freiburg, Germany), and wash solutions were supplied with a multichannel pipette very slowly via the lateral planes of the plate wells. Adherent cells were safranin-stained (2 min with 0.1% safranin; Merck, Darmstadt, Germany) and the absorbance of stained biofilms was measured at 530 nm after resolving the stain with 200 µL 30% (*v/v*) acetic acid, using a microtiter plate reader (EnSight; Perkin Elmer, Rodgau, Germany).

The assessment of biofilm formation on peripheral venous catheter (PVC, Venflon Pro Safety 14 G; BD) fragments under dynamic conditions was carried out as describe in [49]. PVC fragments of 1 cm length were placed into reaction tubes filled with 1 mL of TSB and inoculated with 1 × 10^7^ CFU of PBS-washed bacterial cells obtained from exponential growth phase. The PVC fragments were incubated under non-limited nutrient conditions for five days at 37 °C and 150 rpm, and the media were replaced with fresh media every 24 h. Five days post inoculation, PVC fragments were placed into fresh reaction tubes filled with 1 mL of TSB. The biofilms were detached from the catheter surface and resolved by sonication (50 watt for 5 min) followed by 1 min of vortexing. CFU rates and biomasses of resolved biofilms and culture supernatants at day five post inoculation were determined by plate counting and OD_600_ measurements, respectively.

Biofilm formation was additionally monitored and quantified in a glass tube assay. Briefly, 3 mL pre-warmed TSB were inoculated to an OD_600_ of 0.05 in glass tubes, and cultures were incubated at 37 °C and 150 rpm for 18 h. Biofilms formed at the air-liquid interface and bottom of the glass tubes were rinsed two times with PBS (pH 7.2) and air-dried for one hour. After staining with 0.1% safranin (Merck) for two minutes, glass tubes were rinsed three times with ddH_2_O, and stained biofilms documented photographically. Safranin bound to the biofilm was subsequently released by adding 1 mL of 30% acetic acid. 200 µL of the solution was then transferred to a 96-well plate (flat-bottom; Sarstedt, Nümbrecht, Germany) and the color intensity quantified at an absorbance of 530 nm (*A*_530_).

### 4.6. Cloning, Expression and Purification of Recombinant SpoVG

A synthetic DNA fragment covering the *S. epidermidis* 1457 *spoVG* (B4U56_11365) ORF with tags (ATTB1 recombination site–Shine-Dalgarno sequence–Kozac sequence– Histidine Affinity Tag [HAT]–Human influenza hemagglutinin [HA] tag–linker–spoVG- ATTB1 site) was ordered by IDT (Integrated DNA Technologies Germany GmbH, Munich, Germany). The DNA fragment was integrated in pDONR 221 to generate HAT-HA-SpoVG_pDONR 221. The sequence was recombined into the pRSF RfA Destination vector (MGC Platform; Montpellier, France), thus generating pRSF-HAT-HA-SpoVG harboring an ORF encoding the protein sequence HAT-HA-SpoVG (139 aa, 15.91 kD). *E. coli* BL21 Star cells (Invitrogen) transformed with pRSF-HAT-HA-SpoVG were grown in LB medium containing 1 mg/mL of glucose and 100 µg/mL of ampicillin.

For protein expression, pRSF-HAT-HA-SpoVG harboring BL21 Star cells were grown in LB at 37 °C until the culture reached an OD_600_ of 0.8. Cultures with this OD were supplemented with 0.5 mM isopropyl-β-D-thiogalactopyranoside and grown overnight at 20 °C. Bacterial cells were disrupted in a French pressure cell and centrifuged at 14,000 rpm for 25 min. Purifications were performed using a TALON^®^ metal affinity resin (Takara, Saint-Germain-en-Laye, France) according to manufacturer’s instructions and eluted in 200 mM imidazole, 50 mM Tris-HCl [pH 7.4], 300 mM NaCl, and 10% [*v/v*] glycerol. Purity of the HAT-HA-SpoVG fractions was confirmed by Coomassie-staining of SDS-PAGEs loaded with the fractions.

### 4.7. Electrophoresis Mobility Shift Assay

The *icaA* and *icaR* dsDNA probes corresponding to the 80-bp regions upstream of the *icaA* and *icaR* ORFs (Figure 7a) were prepared by annealing complementary DYE682-labeled primers (Eurofins Genomics, Table 3) in 10 mM Tris–HCl (pH 8.0) and 20 mM NaCl. Mixtures were incubated at 95 °C for 5 min, and allowed to cool down to 25 °C. Indicated amounts of purified recombinant HAT-HA-SpoVG were mixed with 200 fmoles of the dsDNA probes in EMSA buffer (20 mM HEPES [pH 8.0], 10 mM KCl, 2 mM MgCl_2_, 0.1 mM EDTA, 0.1 mM dithiothreitol, 50 μg/mL bovine serum albumin, and 10% glycerol) in 20 μL final volume reactions. After incubation at room temperature for 30 min, the reactions were run on a 2.5% agarose gel in cold TAE (Tris-Acetate-EDTA) buffer. Gels were imaged using the 700 nm channel of the Odyssey Infrared Imaging System (LI-COR).

### 4.8. Murine Foreign Body Infection Model

Animal experiments were performed with approval of the local State Review Board of Saarland, Germany (project identification code 11/2019 [approved 30 April 2019]), and conducted following the national and European guidelines for the ethical and human treatment of animals. Infection of the animals was carried out as described [41], except that implanted catheter fragments were inoculated with 1 × 10^7^ CFU of *S. epidermidis* strains 1457, its ∆*spoVG* mutant and the *cis*-complemented *spoVG* derivative, respectively. For some of the infected catheter fragments (9/48), co-infections with other bacterial species were noticed. Data obtained from these infection sites were excluded from the study.

### 4.9. Cytokine Determinations

Tissue homogenates obtained from excised tissues surrounding the catheter fragments of *S. epidermidis*-infected mice at day 10 post infection were centrifuged at 4 °C and 2500× *g* for 10 min. Concentrations of cytokines were measured in supernatants using ELISAs for IL-6, KC, and TNF-α (R&D Systems, Minneapolis, MN, USA) as described by the manufacturer.

### 4.10. Statistical Analyses

The statistical significance of changes between groups was determined using the Graph-Pad software package Prism 9.3.1. Identified *p*-values < 0.05 were considered statistically significant. Comparisons between groups were analyzed by the Kruskal-Wallis test and Dunn’s multiple comparisons test.

## Figures and Tables

**Figure 1 ijms-23-03255-f001:**
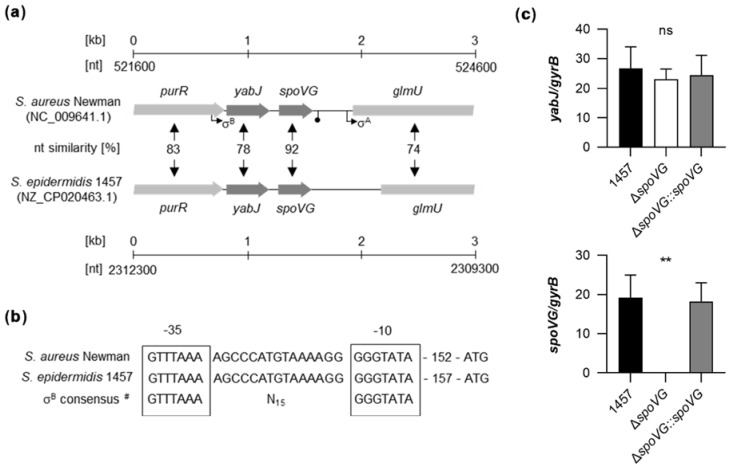
Genetic organization of the *yabJ-spoVG* locus in staphylococci. (**a**) Schematic representation of the *yabJ-spoVG* regions in *S. aureus* Newman and *S. epidermidis* 1457. Open reading frames (ORFs), promoters, terminators, and nucleotide similarities between homologous ORFs are indicated. (**b**) Nucleotide sequences of the σ^B^-promoter regions in front of *yabJ*. σ-factor binding sites are framed. ^#^, σ^B^-promoter consensus motif of *S. aureus* derived from reference [38]. (**c**) Quantitative transcript analyses of *yabJ* and *spoVG* by qRT-PCR in *S. epidermidis* 1457, 1457 Δ*spoVG*, and 1457 Δ*spoVG::spoVG* cells grown in TSB at 37 °C and 225 rpm for 3 h. Transcripts were quantified in reference to the transcription of *gyrB* (in copies per copy of *gyrB*). Data are presented as mean + SD of five biological replicates. **, *p* < 0.01; ns, not significant (Kruskal-Wallis test and Dunn’s multiple comparisons test. Only differences between 1457 and mutants are shown).

**Figure 2 ijms-23-03255-f002:**
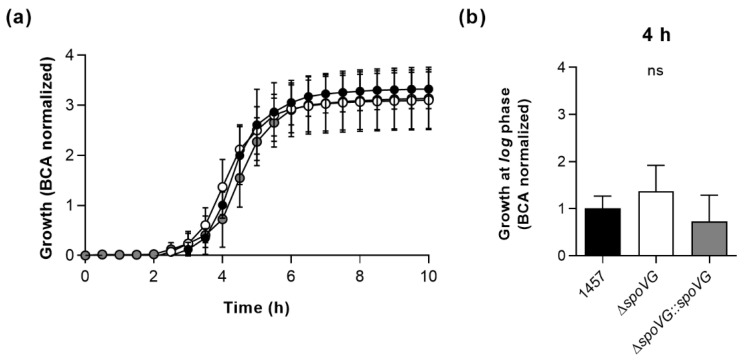
Effect of the *spoVG* deletion on growth of *S. epidermidis* strain 1457 under static conditions in Luria Bertani (LB) broth. Bacterial cells were cultured in LB broth under static conditions at 37 °C and 5% CO_2_ for 10 h. Growth was measured by optical density using the oCelloScope BCA algorithm. (**a**) Growth kinetics of 1457 (black symbols), 1457 Δ*spoVG* (white symbols) and 1457 Δ*spoVG::spoVG* (grey symbols) cell suspensions. The graphs represent the average growth values ± SD of three biological experiments carried out in triplicate. (**b**) BCA normalized optical densities of the cultures at 4 h of growth. ns, not significant (Kruskal-Wallis test and Dunn’s multiple comparisons test).

**Figure 3 ijms-23-03255-f003:**
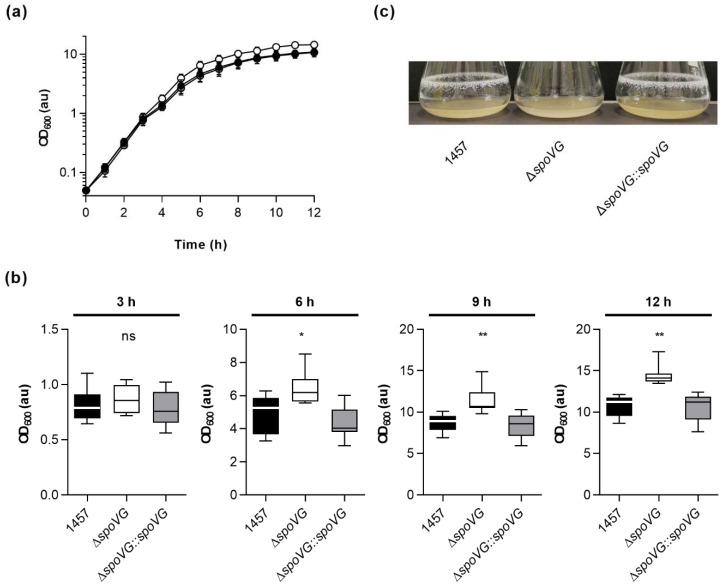
Effect of the *spoVG* deletion on growth of *S. epidermidis* strain 1457 under dynamic conditions in tryptic soy broth (TSB). Bacteria were inoculated to an optical density at 600 nm (OD_600_) of 0.05 in TSB and cultured aerobically at 37 °C and 225 rpm in a culture to flask volume of 1:10. OD_600_ measurements of the cultures were determined hourly. Samples were diluted in PBS when an OD_600_ value of 0.8 was reached. The dilution factor was used to multiply with the measured result for the OD_600_ values displayed in the graphs. Calculated values are given as arbitrary units (au). (**a**) Growth kinetics of 1457 (black symbols), 1457 Δ*spoVG* (white symbols), and 1457 Δ*spoVG::spoVG* (grey symbols) cell suspensions. The results are the mean ± SD of eight biological replicates. (**b**) OD_600_ values (au) of the cell cultures at 3, 6, 9, and 12 h of growth, respectively. Data are presented as box and whisker plot showing the interquartile range (25–75%, box), the median (horizontal line) and the standard deviation (bars) of eight biological replicates. * *p* < 0.05; ** *p* < 0.01; ns, not significant (Kruskal-Wallis test and Dunn’s multiple comparisons test. Only differences between 1457 and mutants are shown). (**c**) Representative images of biofilms formed on glass flasks by cell suspensions cultured in TSB at 37 °C and 225 rpm for 12 h.

**Figure 4 ijms-23-03255-f004:**
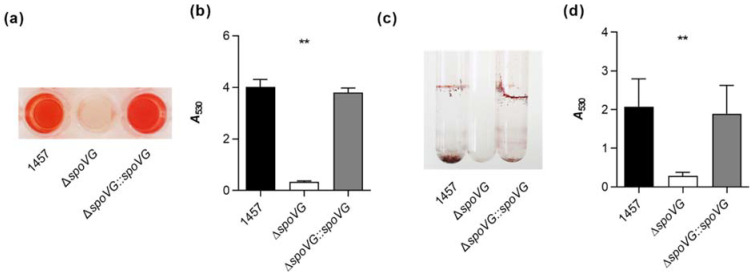
Effect of the *spoVG* deletion on in vitro biofilm formation of *S. epidermidis* strain 1457. (**a**,**b**) Biofilm growth of *S. epidermidis* strains 1457 (black symbols), 1457 Δ*spoVG* (white symbols), and 1457 Δ*spoVG::spoVG* (grey symbols) in a static 96-well microplate assay. Cells were incubated for 18 h at 37 °C in TSB. (**a**) Representative images of safranin-stained biofilms formed at the bottom of the wells. (**b**) Absorption rates of safranin-stained biofilms at 530 nm. The data show the mean + SD of six biological replicates carried out in triplicate. (**c**,**d**) Biofilm growth of the strain triplet under dynamic conditions in glass tubes. (**c**) Representative image of safranin-stained biofilms formed on the glass surfaces after 18 h of growth in TSB at 37 °C and 150 rpm. (**d**) Absorption rates of safranin-stained biofilms at 530 nm. The data show the mean + SD of six biological replicates. ** *p* < 0.01 (Kruskal-Wallis test and Dunn’s multiple comparisons test, only differences between 1457 and mutants are shown).

**Figure 5 ijms-23-03255-f005:**
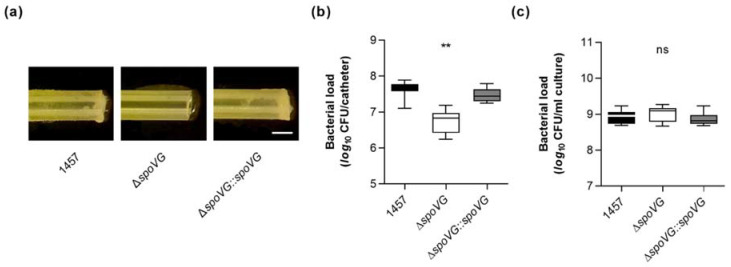
Effect of the *spoVG* deletion on in vitro biofilm formation of *S. epidermidis* strain 1457 on catheter tubing. (**a**) Representative images of *S. epidermidis*-loaded catheter fragments at day 5 post inoculation (12.5-fold magnification). The results are representative of three independent experiments. Scale bar, 1 mm. (**b**,**c**) CFU rates of detached biofilms (**b**) and cultures (**c**) were determined by plate counting. The data are presented as box and whisker plot showing the interquartile range (25–75%, box), the median (horizontal line), and the standard deviation (bars) of nine independent experiments. ** *p* < 0.01; ns, not significant (Kruskal-Wallis test and Dunn’s multiple comparisons test, only differences between 1457 and mutants are shown).

**Figure 6 ijms-23-03255-f006:**
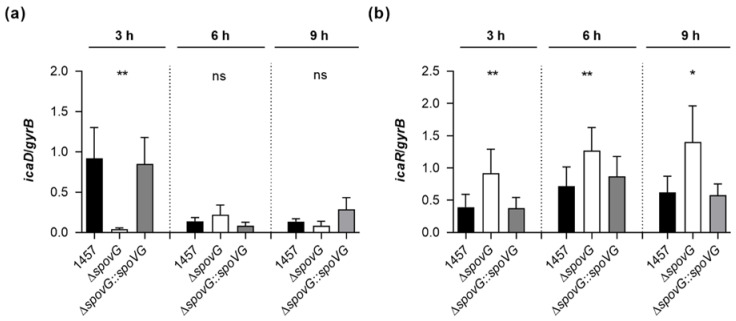
Effect of the *spoVG* deletion on *ica* transcription of *S. epidermidis* strain 1457. Bacterial cells were cultured aerobically in TSB, as outlined in Materials and Methods. Cells were sampled at the time points indicated, total RNAs isolated, and qRT-PCRs performed for *icaD* (**a**) and *icaR* (**b**). Transcripts were quantified in reference to the transcription of gyrase B (*gyrB*). Data are presented as mean + SD of eight biological replicates. * *p* < 0.05; ** *p* < 0.01; ns, not significant (Kruskal-Wallis test and Dunn’s multiple comparisons test, only differences between 1457 and mutants are shown).

**Figure 7 ijms-23-03255-f007:**
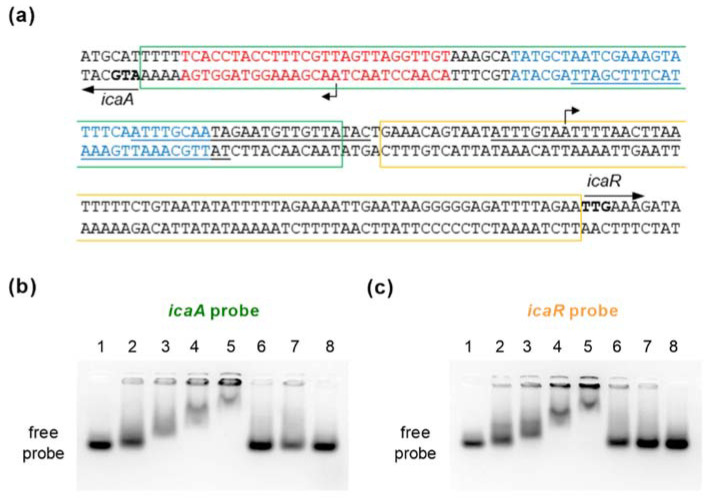
EMSA using purified SpoVG and double-stranded DNA probes generated from the *icaA-icaR* intergenic region. (**a**) Nucleotide sequences of the *icaA-icaR* intergenic region of *S. epidermidis* 1457 (NCBI Reference Sequence NZ_CP020463.1). Start codons of the *icaA* and *icaR* ORFs are highlighted in bold. The highest affinity TcaR binding site proposed by [24] is indicated by blue letters. Putative SarA binding sites proposed by [19] are underlined. The IcaR binding site in front of *icaA* is depicted in red letters [45]. Regions covered by the *icaA* and *icaR* probes used for EMSAs are framed by green and orange rectangles, respectively. TSS of *icaA* [46] and *icaR* [47] are indicated by bent arrows (**b**,**c**) EMSAs were performed using purified recombinant N-terminally HAT-HA-tagged SpoVG and a 80-mer labeled dsDNA fragments covering the *icaA* (**b**) and *icaR* (**c**) TSS, respectively. Controls with no HAT-HA-SpoVG protein added are shown in the first lane, or with 2, 4, 8, and 12 µM of purified HAT-HA-SpoVG, respectively, in lanes 2 to 5. Labeled probes were also tested in presence of an excess of 20× or 10× unlabeled *icaA* (**b**) and *icaR* (**c**) probes (lanes 6 and 7; order switched between (**b**,**c**). A non-related 80-mer labeled dsDNA fragment in combination with 12 µM of HAT-HA-SpoVG served as specificity control (lane 8). Data are representative of at least two independent experiments.

**Figure 8 ijms-23-03255-f008:**
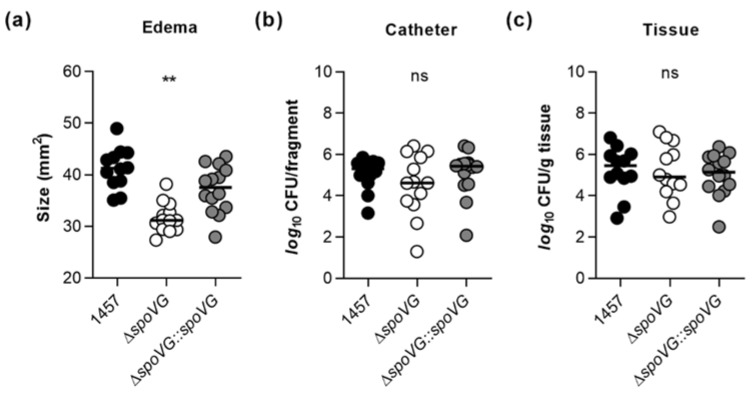
Effect of the *spoVG* deletion on infectivity of *S. epidermidis* strain 1457 in a murine foreign body infection model. Catheter fragments were implanted subcutaneously into the back of normoglycemic mice and inoculated with cells of *S. epidermidis* strains 1457 (black symbols), 1457 Δ*spoVG* (white symbols) and 1457 Δ*spoVG::spoVG* (grey symbols), respectively. (**a**–**c**) Ten days post infection, animals were euthanized, edema sizes around the implanted catheters were measured (**a**), and the catheters and surrounding tissues were explanted. Bacterial loads from catheter detached biofilms (**b**) and in surrounding tissue homogenates (**c**) were determined by CFU counting (*n* = 12–14 per group). The data represent the values of every individual edema/catheter/tissue (symbols) and the median (horizontal line). **, *p* < 0.01; ns, not significant (Kruskal-Wallis test and Dunn’s multiple comparisons test, only differences between 1457 and mutants are shown).

**Figure 9 ijms-23-03255-f009:**
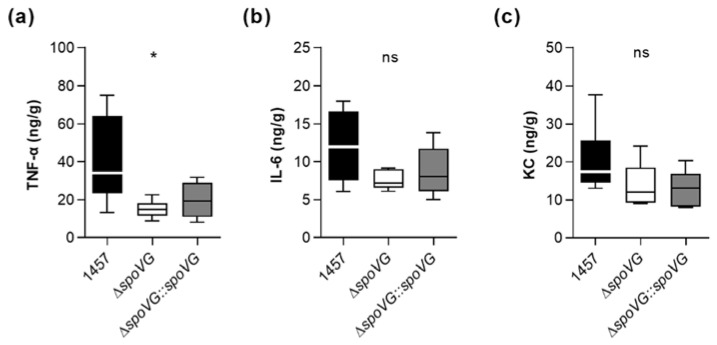
Effect of the *spoVG* deletion on pro-inflammatory cytokine level in *S. epidermidis* 1457 infected tissue. Catheter fragments were implanted subcutaneously into the back of normoglycemic mice and inoculated with cells of *S. epidermidis* strains 1457 (black symbols), 1457 Δ*spoVG* (white symbols), and 1457 Δ*spoVG::spoVG* (grey symbols), respectively. (**a**–**c**) Concentrations of TNF-α (**a**), IL-6 (**b**), and KC (**c**) in tissue homogenates surrounding the infected catheter collected at day 10 post infection (*n* = 6–7 per group). Data are presented as box and whisker plot showing the interquartile range (25–75%, box), the median (horizontal line) and the standard deviation (bars). *, *p* < 0.05; ns, not significant (Kruskal-Wallis test and Dunn’s multiple comparisons test, only differences between 1457 and mutants are shown).

**Table 1 ijms-23-03255-t001:** Generation time of *S. epidermidis* exponential growth phase cells cultivated in TSB under aerobic conditions.

Strain	Generation Time (min) ^1^	*p* Value ^2^
1457	44.69 ± 6.77	
Δ*spoVG*	42.53 ± 1.90	0.999
Δ*spoVG::spoVG*	42.48 ± 5.85	0.610

^1^ Data are presented as mean ± SD (*n* = 8). ^2^
*p* values were determined by Kruskal-Wallis test and Dunn’s multiple comparisons test (only differences between 1457 and mutants are shown).

**Table 2 ijms-23-03255-t002:** Strains and plasmids used in this study.

Strain	Description ^1^	Reference or Source
** *S. aureus* **		
RN4220	NCTC8325-4 derivative, acceptor of foreign DNA	[59]
** *S. epidermidis* **		
1457	Central venous catheter isolate, high level PIA producer	[37]
1457 M10	Biofilm-negative 1457 *icaA*-Tn*917* transposon mutant; Erm^R^	[60]
1457 M15	*rsbU*-Tn*917* mutant of 1457 used as recipient by electroporation; Erm^R^	[56]
1457 Δ*spoVG*	1457 Δ*spoVG*::*ermB*; Erm^R^	This study
1457 Δ*spoVG::spoVG*	*cis*-complemented 1457 Δ*spoVG* derivative	This study
** *E. coli* **		
BL21 Star (DE3)	Protein expression strain	Invitrogen
DH5α	Cloning strain	Invitrogen
IM08B	*E. coli* DC10B derivative harboring *hsdS* of *S. aureus* strain NRS384, Δ*dcm*	[61]
TOP10	*E. coli* derivative ultra-competent cells used for general cloning	Invitrogen
**Plasmids**		
pBASE6	*E. coli–Staphylococcus* temperature-sensitive suicide shuttle vector, *secY* counterselection; *bla cat*	[62]
pBASE6_*spoVG*_comp	pBASE6 derivative harboring the C-terminal region of *purR*, *yabJ-spoVG*, and the N-terminal region of *glmU*	This study
pBT2	Temperature sensitive *E. coli-Staphylococcus* shuttle plasmid; *bla*, *cat*	[63]
pBT2_*spoVG*_KO	pBT2 derivative harboring *spoVG* up-*ermB*-*spoVG* do; *bla, cat, ermB*	This study
pDEST R4-R3	Gateway-destination vector, contains *ccdB* and *att*R4/R3 sites; *bla*, *cat*	Invitrogen
pDEST_*spoVG*_ko	pDEST R4-R3 derivative harboring *spoVG* up-*ermB*-*spoVG* do; *bla, cat, ermB*	This study
pDONR 221	Entry vector to clone *att*B1 and *att*B2 flanked PCR products; *ccdB*+, *cat*	Invitrogen
pDONR P2R-P3	Entry vector to clone *att*B2 and *att*B3 flanked PCR products	Invitrogen
pDONR P4-P1R	Entry vector to clone *att*B4 and *att*B1 flanked PCR products	Invitrogen
pENTRY_*ermB*	pDONR 221 derivative harboring *att*B1 and *att*B2 flanked *ermB*	This study
pENTRY_*spoVG*_up	pDONR P4-P1R derivative harboring *att*B4 and *att*B1 flanked *spoVG* up	This study
pENTRY_*spoVG*_do	pDONR P2R-P3 derivative harboring *att*B2 and *att*B3 flanked *spoVG* do	This study
HAT-HA-EpiSpoVG_pDon221	pDONR 221 derivative harboring ATTB1, Shine-Dalgarno and Kozac sequences, a HAT clontech tag, a HA tag, a linker region and *spoVG*; *cat*	This study
pRSF-HAT-HA-SpoVG	pRSF RfA derivative harboring the *spoVG* ORF N-terminally fused to HAT and HA tags; *kan*	This study
pRSF RfA	Gateway-destination vector; *kan*	Montpellier Genomic Collection (MGC)

^1^ Erm^R^, erythromycin-resistant; PIA, polysaccharide intercellular adhesin.

**Table 3 ijms-23-03255-t003:** Primers used in this study.

Primer	Direction	Sequence (5′-3′) ^1^
Cloning primer		
spoVG up	forward	*GGGGACAACTTTGTATAGAAAAGTTG*TTAAAGTGGAACCAGGCAAC
	reverse	*GGGGACTGCTTTTTTTGACAAACTTG*CTAGCGTAATGGAAACGAGTG
spoVG do	forward	*GGGGACAGCTTTCTTGTACAAAGTGGTC*AGATAACGAAGAATCAGAC
	reverse	*GGGGACAACTTTGTATAATAAAGTTG*GAAAACGAGGTCATCAAACC
*ermB*	forward	*GGGGACAAGTTTGTACAAAAAAGCAG*GCTGACGGTGACATCTCTCTATTG
	reverse	*GGGGACCACTTTGTACAAGAAAGCTGGGTG*AAAAGGTACCATAAACGGTCG
*spoVG*_KO	forward	atggtactgcagTTAAAGTGGAACCAGGCAAC
	reverse	atggtaggatccGAAAACGAGGTCATCAAACC
MBH603	forward	gtcgagctCATGCTACAATGTGGTGCTG
MBH607	reverse	gacgagaTCTTTAATCGCACGTTCTGAGTC
**Verification primer**		
erm_v	forward	AATTGGAACAGGTAAAGGGC
spoVG_do_v	reverse	GGTTTGATAATTTTAGAAATTC
MBH608	forward	GGTACCCTAAGCACTAGGCCCATATTCGC
**qRT-PCR primer**		
*gyrB*	forward	CTAATGCTGATTTACGACGCGTAA
	reverse	TCTGTAGGACGCATTATTGTTGAAA
*icaD*	forward	GTATTGTATCGTTGTGATGAT
	reverse	ACTTTCCATTTGAGAATTGAT
*icaR*	forward	ATGGTACTACACTTGATGATA
	reverse	GTAATGATAATATAGACTAGCCTTT
*yabJ*	forward	AAAGCGACAATCTATATTTCTGA
	reverse	ACCTATCAATTCAATTTCTACCTT
*spoVG*	forward	GCAGTGATGAAAGTATATGATGA
	reverse	CTTCGTCTGATTCTTCGTTATC
**EMSA primer**		
*icaA* p	forward	TAACAACATTCTATTGCAAATTGAAATACTTTCGATTAGCATAT-GCTTTACAACCTAACTAACGAAAGGTAGGTGAAAAA
	reverse	TTTTTCACCTACCTTTCGTTAGTTAGGTTGTAAAGCATATGCTAA-TCGAAAGTATTTCAATTTGCAATAGAATGTTGTTA
*icaR* p	forward	GAAACAGTAATATTTGTAATTTTAACTTAATTTTTCTGTAATAT-ATTTTTAGAAAATTGAATAAGGGGGAGATTTTAGAA
	reverse	TTCTAAAATCTCCCCCTTATTCAATTTTCTAAAAATATATTACA-GAAAAATAAGTTAAAATTACAAATATTACTGTTTC
*secA* p	forward	AGTATAATTTTCTAACTATAAATGATAAGATATATTGTTGTAG-GCCAAACAGTTTTTTAGCTAAAGGAGCGAACGAAATG
	reverse	CATTTCGTTCGCTCCTTTAGCTAAAAAACTGTTTGGCCTACAA- CAATATATCTTATCATTTATAGTTAGAAAATTATACT

^1^ Italic nucleotides indicate attachment sites for Gateway cloning. Small letters indicate nucleotides that do not fit with the target DNA. Restrictions sites used for cloning are underlined.

## Data Availability

The datasets generated and analyzed during the current study are available from the corresponding author on reasonable request.

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
