# Peer review of "The Transcription Factor SpoVG Is of Major Importance for Biofilm Formation of Staphylococcus epidermidis under In Vitro Conditions, but Dispensable for In Vivo Biofilm Formation"

_ijms, 2022, doi:10.3390/ijms23063255_

Round 1

Reviewer 1 Report

Comments for Authors:

The manuscript presents a thorough experimental study for the role of the SpoVG transcription factor in S. epidermidis biofilm formation.

Please find detailed comments below:

  1. Page 4, Figure 2:

“Effect of the spoVG deletion on growth of S. epidermidis strain 1457 under static conditions in Luria Bertani (LB) broth”. The culture was only 10 hours old; wouldn’t biofilm formation require longer growth periods?

  1. Why are different time periods used for culturing in the different experiments? Figure 2: 10 hours, Figure 3: 12 hours, Figure 4: 18 hours, Figure 5: five days.

  1. How was it ensured that the biofilm doesn’t get washed off during the removal of the planktonic cells?

  1. Page 5, Figure 3b:

Are the ordinates (y axes) showing OD600 values? How can they be as high as 20? For most of the spectrophotometric samples, O.D. should be 1 or less than 1 but it can be up to 3. If a sample has O.D. greater than 3 this means only 1 photon out of 1000 will be detected by the detector. Please explain.

  1. Page 7, Figure 5: Was the same PVC catheter tubing material used later in the murine experiments? Could the different environment (culture conditions vs murine physiology) account for the differences between in vivo and in vitro experimental results? Would it be possible to provide the same conditions in the in vitro setup?

Also, Figure 8 seems to have a significantly lower inoculum (log 5 CFU/fragment or CFU/g tissue) compared to the in vitro testing on Figure 5 (log 8 CFU/catheter).

Author Response

Comments for Authors:

The manuscript presents a thorough experimental study for the role of the SpoVG transcription factor in S. epidermidis biofilm formation.

We thank this reviewer for his/her positive response and comments.

Please find detailed comments below:

1. Page 4, Figure 2: “Effect of the spoVG deletion on growth of S. epidermidis strain 1457 under static conditions in Luria Bertani (LB) broth”. The culture was only 10 hours old; wouldn’t biofilm formation require longer growth periods?

With the experiments shown in Figure 2 we intended to monitor the growth kinetics of S. epidermids cells under static conditions over time and did not focus on biofilm formation. However, given that bacteria were cultured under static conditions, we have to expect that S. epidermidis cells sedimented more and more over time and accumulated on the bottom of the plate. This might be also the reason for our findings that no further increase in BCA values were seen after 6-8 h of cultivation in this system, as the bottoms of the wells might be completely covered by bacterial cells at this time point.

2. Why are different time periods used for culturing in the different experiments? Figure 2: 10 hours, Figure 3: 12 hours, Figure 4: 18 hours, Figure 5: five days.

In the experiments shown in Figure 2, the in vitro growth kinetics of S. epidermidis 1457 and its spoVG mutants were monitored. In our pilot experiments, we noticed that a steady state of the BCA normalized optical densities was reached already after 6-8 h of cultivation, so we focused on the first 10 h of growth in this assay.

In our dynamic growth experiments shown in Figure 3, we extended our observation window to 12 h, as we still observed a relevant increase in OD600 during the period 10 to 12 h.

The period used for the experimental results given in Figure 4 was adopted from previous studies (our reference 40 of the revised version). We have to admit that we never checked for biofilm formation of S. epidermidis in the static 96 well-based biofilm model after 12 h of growth. However, we
visually inspected the biofilm formation of S. epidermidis cultured under dynamic conditions in glass tubes for 12 h, which was comparable to the biofilm formed on the glass tubing after 18 h of incubation, indicating that the biomass of the biofilm is not markedly increased by extending the cultivation time from 12 to 18 h, probably because most of the nutrients of the culture medium are already spent after 12 h of growth.

Biofilm formation on catheter tubing requested a longer observation period, as no macroscopically visible biofilm was seen after 1 or 2 days of incubation. First macroscopic signs of a biofilm were seen after 3 days of incubation and further increased until day 5. 

3. How was it ensured that the biofilm doesn’t get washed off during the removal of the planktonic cells?

We assume that this reviewer is referring to our results gained with the static biofilm model, and he/she is completely right that this is a weak point of this assay. In fact, we cannot exclude that the spoVG mutant formed a weakly bound biofilm, which was detached from the surface of the well plate bottom by removing the culture medium and/or washing the wells with PBS. We tried to introduce as little sheer stress as possible to the biofilms by not pipetting the liquids directly onto the biofilms and by using a microchannel pipette (addition of liquids) and glass capillary attached to a vacuum pump system (removal of liquids) to introduce comparable forces to the biofilms. This information is now given in the methods part of the revised version (lanes 507-510).

For the catheter related in vitro biofilm model, we can only tell that removal of the media in the lumens of the catheter tubing did not alter the biofilms formed on the outside of the tubing, but we cannot exclude that our washing step removed at least some of the biofilm biomass formed on the inside of the catheter tubing.  

4. Page 5, Figure 3b: Are the ordinates (y axes) showing OD600 values? How can they be as high as 20? For most of the spectrophotometric samples, O.D. should be 1 or less than 1 but it can be up to 3. If a sample has O.D. greater than 3 this means only 1 photon out of 1000 will be detected by the detector. Please explain.

For spectrophotometric OD600 measurements, we routinely dilute our culture samples in PBS if the culture OD at 600 nm reaches a value above 0.8 to assure that the spectrophotometric OD600 readings of the diluted culture samples are always in the linear range (i.e. below 0.8; see Eppendorf WHITE PAPER No. 28; https://www.eppendorf.com/product-media/doc/en/148370/Detection_White-Paper_028_BioPhotometer-D30_BioSpectrometer-family_OD600-Measurements-Different-Photometers.pdf). That way, TSB grown S. epidermidis cultures can reach OD600 values close to 20 with the setup used in our laboratory. The info about the dilution step is now given in the Materials part of this revised version (lanes 444-447).  

5. Page 7, Figure 5: Was the same PVC catheter tubing material used later in the murine experiments? Could the different environment (culture conditions vs murine physiology) account for the differences between in vivo and in vitro experimental results? Would it be possible to provide the same conditions in the in vitro setup?

Yes, we used the same catheter tubing in our in vitro and in vivo assays. We also assume that the different physiological conditions encountered by S. epidermidis in our in vitro and in vivo PVC tubing assays might be the reason why we see such differences between both assays. Further support for this hypothesis is given by a recent publication showing that PIA and/or embp-negative S. epidermidis failed to form biofilms in in vitro assays but were able to do so, if the culture medium was supplemented with human plasma or serum (DOI: 10.1099/jmm.0.001287). For the future, we will update our in vitro PVC assay by adding human plasma to the growth medium. We will also try to culture S. epidermidis contaminated PVC tubing in heparinized whole blood, however, this requires an ethical vote first. If the latter strategy is successful, we approach the in vivo situation more and more. A further improvement could be the application of sheer stress to the system; however, this would require large volumes of human blood and thus be in conflict with ethical issues, as blood is needed for clinical interventions probably all over the world. 

Also, Figure 8 seems to have a significantly lower inoculum (log 5 CFU/fragment or CFU/g tissue) compared to the in vitro testing on Figure 5 (log 8 CFU/catheter).

We also assume that the differences in biomasses seen in our in vitro PVC assay and in the in vivo model are probably due to the impact of the host immune system, which is present in the in vivo system but absent in our in vitro PVC fragment model. Additionally, differences in CFU numbers might be driven by nutrient supply, which might have been higher in our in vitro test system, in which growing biofilms were supplemented with fresh medium on a daily basis.

Reviewer 2 Report

Please follow the instructions to improve the quality.

Manuscript details:

Manuscript ID: ijms-1623917
Type of manuscript: Article
Title: The transcription factor SpoVG is of major importance for biofilm formation of Staphylococcus epidermidis under in vitro conditions, but dispensable for in vivo biofilm formation

General Comments: The author explored to transcription factor SpoVG is essential for the capacity of S. epidermidis to form such biofilms on artificial surfaces under in vitro conditions.

This reviewer perused this manuscript with great interest and pleasure, and truly believes in its scientific contribution to the field of Science. They also pointed out the deletion of spoVG significantly altered the expression of the intercellular adhesion (ica) locus by upregulating the transcription of 31 the ica operon repressor icaR and down-regulating the transcription of icaADBC. However, this reviewer enlisted some observation regarding the article.

First and foremost: The manuscript is well-written, with particular attention to English spelling, grammar, syntax, and semantics, as well as scientific style. The authors should correct the remarks below before accepting the manuscript by editors.                                    

Specific comments:

Comment 1: In line 38, please mention full form first time PIA-dependent (for the new reader).

Comment 2: In line 50, first time full name then you may write S. epidermidis.

Comment 3: Please check the Table 3 (format style).

Comment 4: If possible please update the references (very old).

Comment 5: Please update the introduction with recent references.

Author Response

General Comments: The author explored to transcription factor SpoVG is essential for the capacity of S. epidermidis to form such biofilms on artificial surfaces under in vitro conditions.

This reviewer perused this manuscript with great interest and pleasure, and truly believes in its scientific contribution to the field of Science. They also pointed out the deletion of spoVG significantly altered the expression of the intercellular adhesion (ica) locus by upregulating the transcription of 31 the ica operon repressor icaR and down-regulating the transcription of icaADBC. However, this reviewer enlisted some observation regarding the article.

First and foremost: The manuscript is well-written, with particular attention to English spelling, grammar, syntax, and semantics, as well as scientific style. The authors should correct the remarks below before accepting the manuscript by editors.     

We thank this reviewer for his/her positive response and remarks.                            

Specific comments:

Comment 1: In line 38, please mention full form first time PIA-dependent (for the new reader).

We introduce the full form for PIA in lane 25 of the revised version. "Inactivation of spoVG in the polysaccharide intercellular adhesin (PIA) producing S. epidermidis strain 1457...."

Comment 2: In line 50, first time full name then you may write S. epidermidis.

"Staphylococcus aureus and Staphylococcus epidermidis" was changed to "Staphylococcus aureus and S. epidermidis" in the revised version.

Comment 3: Please check the Table 3 (format style).

We adapted the format style of Table 3.

Comment 4: If possible please update the references (very old).

We agree with this reviewer that some of our references are very old. We intended to acknowledge the publications indicating a certain finding for the first time to our knowledge, and we hope that this reviewer can agree to this procedure.  

Comment 5: Please update the introduction with recent references.

We updated the literature in the introduction by citing the most recent reviews in this field and added new findings related to this topic. If we missed any recent literature that should be cited here, we would greatly appreciate any advice.  

Round 2

Reviewer 1 Report

This Reviewer would like to thank the Authors for adequately addressing most of the review comments. Regarding Fig. 3b, the multiplication of the OD600 value with the dilution factor, it may be acceptable at lower concentrations where linear correlation exists, however, an OD600 value shouldn't be over 3 or 4. In the Eppendorf White Paper example the dilution factor was applied in a way that the product/result stays in the quasi-linear range that is  physically measurable. To extrapolate into unmeasurable regions (OD600 > 20) where no linear correlation would exist cannot be supported by valid data. Please include this reasoning carefully in the manuscript.

Author Response

We thank this reviewer for raising our awareness for this point. We adapted our Figure 3 and indicate our calculated OD600 values now as arbitrary units and explain in the legend to this Figure, how these values were obtained.